# Bioactive Constituents from the Roots of *Eurycoma longifolia*

**DOI:** 10.3390/molecules24173157

**Published:** 2019-08-30

**Authors:** Jingya Ruan, Zheng Li, Ying Zhang, Yue Chen, Mengyang Liu, Lifeng Han, Yi Zhang, Tao Wang

**Affiliations:** 1Tianjin State Key Laboratory of Modern Chinese Medicine, 312 Anshanxi Road, Nankai District, Tianjin 300193, China; 2Tianjin Key Laboratory of TCM Chemistry and Analysis, Institute of Traditional Chinese Medicine, Tianjin University of Traditional Chinese Medicine, 312 Anshanxi Road, Nankai District, Tianjin 300193, China

**Keywords:** *Eurycoma longifolia* roots, eurylophenolosides, eurylolignanosides, piscidinol A, 24-*epi*-piscidinol A, bourjotinolone A, scopoletin, RAW 246.7 cell, anti-inflammatory activity

## Abstract

Four new phenolic components, eurylophenolosides A (**1**) and B (**2**), eurylolignanosides A (**3**) and B (**4**), along with twelve known compounds were isolated from the roots of *Eurycoma longifolia* Jack. The structure of these components was elucidated by using various spectral techniques and chemical reactions. Among the known isolates, syringaldehyde (**12**), 3-chloro-4-hydroxybenzoic acid (**13**), 3-chloro-4-hydroxyl benzoic acid-4-*O*-β-d-glucopyranoside (**14**), and isotachioside (**15**) were isolated from the *Eurycoma* genus for the first time. Further, the NMR data of **14** was reported here firstly. Meanwhile, the nitric oxide (NO) inhibitory activities of all compounds were examined in lipopolysaccharide (LPS)-stimulated RAW264.7 cells at 40 μM. As results, piscidinol A (**6**), 24-*epi*-piscidinol A (**7**), bourjotinolone A (**10**), and scopoletin (**16**) were found to play important role in suppressing NO levels without cytotoxicity. Furthermore, the Western blot method was used to investigate the mechanism of compounds **6**, **7**, **10**, and **16** by analysing the level of inflammation related proteins, such as inducible nitric oxide synthase (iNOS), interleukin-6 (IL-6), and nuclear factor kappa-light-chain-enhancer of activated B cells (NF-κB) in LPS-stimulated RAW264.7 cells. Consequently, compounds **6**, **7**, **10**, and **16** were found to significantly inhibit LPS-induced protein expression of IL-6, NF-κB and iNOS in NF-κB signaling pathway. Moreover, it was found that the protein expression inhibitory effects of **6**, **7**, and **16** exhibited in a dose-dependent manner. The mechanism may be related to the inhibition of the iNOS expressions through suppressing the IL-6-induced NF-κB pathway.

## 1. Introduction

As we know, inflammation is a very common and important pathological process. Although moderate inflammation is a benefit for the body, overreaction or a lack of it can cause some adverse reactions, and even lead to other deadly chronic disease such as cancer, Alzheimer’s disease, diabetes, and atherosclerosis [1]. Thus, the discovery of anti-inflammatory drugs and the treatment of inflammation are particularly essential.

As a kind of immune cell distributed throughout the body, macrophages play a central role in the immune surveillance system [2]. Macrophages can immediately play an immune role indirectly by releasing various inflammatory agents after the pathogen enters the body. As one type of the macrophage-like cell line, the RAW 264.7 cell is a common cell line for studying microbiological immunology and other related research fields because of its strong ability to adhere to phagocytosis antigens [3].

Inflammatory response is closely related to inflammatory factors and inflammatory cells. Proinflammatory cytokines regulate the secretion of inflammatory factors. Cellular signalling pathways can regulate the transcription and synthesis of pro-inflammatory cytokines, which include Janus kinase-signal transduction and transcription activator (JAK-STAT), mitogen-activated protein kinase (MAPK), and nuclear factor kappa-light-chain-enhancer of activated B (NF-κB) pathways. Among them, NF-κB is the main signaling pathway [4].

Lipopolysaccharides (LPS) can stimulate the acute inflammatory response of RAW 264.7 cells to release typical proinflammatory cytokines, such as tumor necrosis factor (TNF-α) and interleukin 6 (IL-6) [1]. Then, protein kinase, NF-κB is activated. After this, the level of inducible nitric oxide synthase (iNOS) in an abnormal body is regulated to promote the synthesis of nitric oxide (NO), and then the expression of cyclooxygenase-2 (COX-2) is upregulated. The process will promote tissue damage and chronic disease.

*Eurycoma longifolia* Jack (Simaroubaceae family) is a wild shrub. As a commonly used medicine in Southeast Asian countries, it is mainly distributed in Malaysia, Vietnam, Thailand, and India. It is also known as one of the three national treasures of Malaysia, together with bird’s nest and tin. Its roots and root bark possess multiple biological functions such as male testosterone level increasement, anti-fatigue, hypertension, fever treatment [5,6], and so on. The main constituents in it are quassinoids, alkaloids, and terpenoids [7]. Pharmacological studies have shown that *E. longifolia* exhibited anti-malarial, anti-cancer, anti-inflammatory, and other effects [8,9,10,11]. Among them, anti-cancer and anti-malarial activities are research hotspots. However, the study of its anti-inflammatory activity is rare [12,13].

In this paper, the phytochemistry and anti-inflammation therapeutic substance in *E. longifolia* roots were investigated. Chromatographies and spectral analysis techniques were combined to isolate and identify constituents from the plant. The inhibitory activities of all gained compounds against NO production in RAW 264.7 cells induced by LPS were evaluated. Furthermore, the anti-inflammatory mechanism of potential activity compounds was studied using the Western blot method.

## 2. Results and Discussion

The 70% EtOH extract of *E. longifolia* roots was suspended in water and partitioned with EtOAc. The H_2_O soluble extract was subjected to D101 macroporous resin column chromatography (CC), and eluated with H_2_O and 95% EtOH, successively. Separation of the EtOAc fraction and 95% EtOH eluated fraction by column chromatography (CC) such as silica gel, Sephadex LH-20, and preparative high-performance liquid chromatography (pHPLC) yielded four new phenolic components, namely eurylophenolosides A (**1**) and B (**2**), eurylolignanosides A (**3**) and B (**4**) (Figure 1). The structures of them were elucidated by using various spectral techniques (^1^H and ^13^C NMR, ^1^H ^1^H COSY, HSQC, HMBC, UV, IR, MS, [α]_D_) and chemical reaction. Moreover, the twelve known isolates, hispidol B (**5**) [14,15], piscidinol A (**6**) [14], 24-*epi*-piscidinol A (**7**) [16], bourjotinolone B (**8**) [17], 3-episapeline A (**9**) [15], bourjotinolone A (**10**) [15], 3-methoxy-4-hydroxybenzoic acid (**11**) [18], syringaldehyde (**12**) [19,20], 3-chloro-4-hydroxybenzoic acid (**13**) [21], 3-chloro-4-hydroxyl benzoic acid-4-*O*-β-d-glucopyranoside (**14**), isotachioside (**15**) [22], and scopoletin (**16**) [23] (Figure 2) were identified by comparing the spectroscopic data with those reported in the corresponding literatures. Among the known compounds, **12**–**15** were isolated from the *Eurycoma* genus for the first time. And the NMR data of **14** was reported here firstly.

Eurylophenoloside A (**1**) was obtained as a white powder with negative optical rotation ([α]_D_^25^ −79.0, MeOH). Its molecular formula, C_25_H_38_O_17_, with seven degrees of unsaturation, was deduced from the quasimolecular ion peak at *m/z* 633.2008 [M + Na]^+^ (calcd for C_25_H_38_O_17_Na, 633.2001) in the HRESI-TOF-MS. Infrared (IR) spectrum of it showed characteristic absorptions of hydroxyl (3374 cm^−1^), aromatic ring (1601, 1505, 1462 cm^−1^), and glycosyl bond (1067 cm^−1^). Its ^1^H and ^13^C NMR spectra (Table 1) showed signals attributed to one symmetrical 1,3,4,5-tetra-substituted phenyl [δ 6.94 (2H, s, H-2,6)], three methoxyls [δ 3.79 (3H, s, 4-OCH_3_), 3.88 (6H, s, 3,5-OCH_3_)]. The long-range correlation observations from δ_H_ 6.94 (H-2,6) to δ_C_ 134.3 (C-4), 154.4 (C-3,5), 155.5 (C-1); δ_H_ 3.88 (3,5-OCH_3_) to δ_C_ 154.4 (C-3,5); δ_H_ 3.79 (4-OCH_3_) to δ_C_ 134.3 (C-4) showed in HMBC spectrum (Figure 3) suggested the aglycon of **1** was 3,4,5-trimethoxyphenol. A total of 25 signals were displayed in its ^13^C NMR spectrum, except for the nine ones belonging to 3,4,5-trimethoxyphenol moiety, there were another sixteen carbon signals. Combining with three signals of anomeric protons [δ 5.38 (1H, d, *J* = 7.8 Hz, H-1’), 5.66 (1H, d, *J* = 2.4 Hz, H-1’’’), 6.60 (1H, br. s, H-1’’)] displayed in its ^1^H NMR spectrum, the existences of one hexose and two pentoses were speculated. d-glucose was obtained when compound **1** was hydrolysed with 1 M HCl, which was identified by retention time and optical rotation using chiral detection by HPLC analysis [24]. Furtherly, it was clarified to be β-d-glucopyranosyl since the coupling constant of anomeric proton was 7.8 Hz. The correlation from δ_H_ 5.38 (H-1’) to δ_C_ 155.5 (C-1) suggested the β-d-glucopyranosyl linked with C-1 position of aglycon. Moreover, the long-range correlations from δ_H_ 4.50 (H-2’) to δ_C_ 110.5 (C-1’’); δ_H_ 5.66 (H-1’’’) to δ_C_ 69.0 (C-6’) indicated both 1- and 6-positions of β-d-glucopyranosyl were substituted by pentose group. The chemical shifts of H-2 of two pentoses were assigned according to the proton and proton correlations between δ_H_ 6.60 (H-1’’) and δ_H_ 4.77 (H-2’’); δ_H_ 5.66 (H-1’’’) and δ_H_ 4.69 (H-2’’’). Furthermore, the linkage positions of them were elucidated by the correlations fom δ_H_ 6.60 (H-1’’) to δ_C_ 81.0 (C-3’’); δ_H_ 4.25, 4.28 (H_2_-5’’) to δ_C_ 75.9 (C-4’’), 81.0 (C-3’’); δ_H_ 5.66 (H-1’’’) to δ_C_ 80.3 (C-3’’’); δ_H_ 4.09, 4.14 (H_2_-5’’’) to δ_C_ 74.9 (C-4’’’), 80.3 (C-3’’’) observed in its HMBC spectrum. Finally, both of the two pentoses were identified as β-d-apiofuranosyl by using the method as following. According to the coupling constants of anomeric protons (^3^*J*_1,2_ < 4 Hz) and the trends in ^13^C NMR data of two pentoses, we could speculate them were β-d- or α-d-apiofuranosyl [25]. Finally, both of them were clarified to be β-d-apiofuranosyl by the NOESY experiment. The NOE correlations were observed between δ_H_ 4.77 (H-2’’) and δ_H_ 4.25, 4.28 (H_2_-5’’); δ_H_ 4.69 (H-2’’’) and δ_H_ 4.09, 4.14 (H_2_-5’’’) in the NOESY spectrum. Therefore, the structure of **1** was determined, which was named eurylophenoloside A.

Eurylophenoloside B (**2**) was isolated as a white powder, too. Its molecular formula, C_34_H_46_O_21_, was established by positive-ion HRESI-TOF-MS [*m/z* 813.2438 [M + H]^+^ (calcd for C_34_H_47_O_21_, 813.2424)]. The IR spectrum of it not only presented the characteristic absorptions of hydroxyl (3357 cm^−1^), aromatic ring (1603, 1506, 1462 cm^−1^), glycosyl bond (1067 cm^–1^), but also the characteristic absorption of unsaturated carboxyl (1704 cm^−1^). Comparing its ^1^H and ^13^C NMR spectra (Table 2) with those of **1**, we found that the chemical shift value of C-6’’’ was increased by 2.9, but that of C-5’’’ was reduced by 1.4 in **2**. Then, C-6’” was supposed to be replaced by an acyl group. Moreover, there were two more symmetrical methoxyl [δ 3.77 (6H, s, 3’’’’,5’’’’-OCH_3_)], two more symmetrical aromatic proton [δ 7.68 (2H, s, H-2’’’’,6’’’’)], as well as one more carbonyl [δ_C_ 166.7 (C-7’’’’)] signals displayed in the NMR spectra of **2** than **1**. The presence of them were clarified by the long-range correlations from δ_H_ 7.68 (H-2’’’’,6’’’’) to δ_C_ 120.1 (C-1’’’’), 143.0 (C-4’’’’), 148.7 (C-3’’’’,5’’’’), 166.7 (C-7’’’’); δ_H_ 3.77 (3’’’’,5’’’’-OCH_3_) to δ_C_ 148.7 (C-3’’’’,5’’’’) observed in its HMBC spectrum, and the moiety was decuced to be 3,5-dimethoxy-4-hydroxybenzoyl. Finally, the correlation from δ_H_ 4.86, 4.90 (H_2_-5’’’) to δ_C_ 166.7 (C-7’’’’) indicated that 3,5-dimethoxy-4-hydroxybenzoyl connected with C-5’’’ of compound **1**, and eurylophenoloside B (**2**) was formed.

Eurylolignanoside A (**3**) was obtained as a white powder with negative optical rotation ([α]_D_^25^ −47.2, MeOH). HRESI-TOF-MS determination result [*m/z* 707.2521 [M + Na]^+^ (calcd for C_32_H_44_O_16_Na, 707.2522)] revealed its molecular formula was C_32_H_44_O_16_. d-glucose was analyzed from its acid hydrolysis product [24]. Its ^1^H and ^13^C NMR spectra (Table 3) indicated the existence of one ABX spin coupling systematic phenyl [δ 6.48 (1H, dd, *J* = 1.5, 8.0 Hz, H-6), 6.56 (1H, d, *J* = 8.0 Hz, H-5), 6.57 (1H, d, *J* = 1.5 Hz, H-2)], one 1,3,4,5-tetrasubstituted phenyl [δ 6.95 (1H, d, *J* = 2.0, H-2’), 6.92 (1H, d, *J* =2.0 Hz, H-6’)], two β-d-glucopyranosyls [δ 4.37 (1H, d, *J* = 8.0 Hz, H-1’’’), 4.68 (1H, d, *J* = 7.5 Hz, H-1’’)], along with two methoxyls [δ 3.69 (3H, s, 3-OCH_3_), 3.83 (3H, s, 3’-OCH_3_)]. In addition to the 26 signals represented by the above mentioned moieties and functional groups, there were six more carbon signals displaying in its ^13^C NMR spectrum, which suggested compound **3** was one of phenylpropane glycoside. The correlations between δ_H_ 3.97 (H-8) and δ_H_ 2.73, 2.96 (H_2_-7), 3.69, 3.76 (H_2_-9) showed in the ^1^H ^1^H COSY spectrum suggested the presence of “-CH_2_-CH-CH_2_O-” moiety. Moreover, the exitence of “-CH=CH-CH_2_OH” moiety was clarified by the correlations between δ_H_ 6.30 (H-8’) and δ_H_ 4.33, 4.52 (H_2_-9’), 6.65 (H-7’). On the other hand, the coupling constant (*J* = 16.0 Hz) between H-7’ and H-8’ indicated that the two protons presented *trans* configuration. The planar structure was clarified by the long-range correlations observed from the followling proton to carbon pairs: δ_H_ 3.69 (3-OCH_3_), 6.56 (H-5) to δ_C_ 148.4 (C-3); δ_H_ 6.48 (H-6), 6.57 (H-2) to δ_C_ 145.4 (C-4); δ_H_ 2.73, 2.96 (H_2_-7) to δ_C_ 113.8 (C-2), 122.6 (C-6), 133.2 (C-1), 139.0 (C-5’); δ_H_ 3.97 (H-8) to δ_C_ 119.4 (C-6’), 133.2 (C-1), 139.0 (C-5’), 145.2 (C-4’); δ_H_ 6.92 (H-6’), 6.95 (H-2’) to δ_C_ 145.2 (C-4’); δ_H_ 3.83 (3’-OCH_3_), 6.95 (H-2’) to δ_C_ 153.5 (C-3’); δ_H_ 6.65 (H-7’) to δ_C_ 109.2 (C-2’), 119.4 (C-6’), 135.2 (C-1’); δ_H_ 4.68 (H-1’’) to δ_C_ 145.2 (C-4’); δ_H_ 4.37 (H-1’’’) to δ_C_ 70.8 (C-9’) (Figure 4). The chemical shift value of C-8 (δ_C_ 42.9, in CD_3_OD) suggested the absolute configuration of it might be 8*R* [26]. Finally, it was clarified by its cotton effect [mdeg −23.1 (259 nm)] displayed in circular dichroism (CD) spectrum [27]. Thus, the structure of eurylolignanoside A (**3**) was elucidated.

Eurylolignanoside B (**4**) was a negative optical active ([α]_D_^25^ −42.5, MeOH) white powder. Its molecular formula, C_24_H_30_O_12_ (*m/z* 509.1666 [M − H]^−^; calcd for C_24_H_29_O_12_, 509.1666) was deduced by HRESI-TOF-MS analysis. The ^1^H and ^13^C NMR (Table 4) as well as various 2D NMR spectra (^1^H ^1^H COSY, HSQC, HMBC) suggested its structure was very similar to that of **3**, except that the signals due to one *trans*-hydroxypropenyl and one β-d-glucopyranosyl disappeared, while the signals belonging to one carboxyl [δ_C_ 170.4 (C-7’)] appeared. The substitute position of carboxyl was clarified by the long-range correlations from δ_H_ 7.49 (H-2’) to δ_C_ 122.6 (C-6’), 148.8 (C-4’), 153.2 (C-3’), 170.4 (C-7’) (Figure 4). Moreover, the existence of d-glucose was elucidated by the HCl hydrolysis result [24]. The absolute configuration of C-8 was determined as *R* by using the same method [26,27] as those for compound **3**, and the structure of **4** was elucidated and named as eurylolignanoside B.

By comparing the spectroscopic data with those reported in literature, the known compounds **5**–**16** were identified.

NO is a signaling factor implicated in a variety of inflammatory conditions. Agents that block NO production might be beneficial for the treatment of inflammatory responses. In order to clarify the anti-inflammatory effects of **1**–**16**, the effects of compounds **1**–**16** on LPS-stimulated NO release were measured through the Griess reaction in RAW264.7 cells [28].

Before the experiment, a dimethyl thiazolyl diphenyl tetrazolium (MTT) assay was used to test the cytotoxicities of **1**–**16**. It was found that all of them displayed no significant cytotoxicity at 40 μM concentration (Appendix A). Then, under this concentration, in vitro potential anti-inflammatory effects of all isolates were investigated. As a result, compound **6** exhibited singnificant inhibitoty effects of the NO release, and compounds **7**, **10** and **16** showed moderate inhibitoty activities of the NO production (Table 5).

The summary of anti-inflammatory activity of triterpenoids suggested that 3-C=O was the key group for the anti-inflammatory activity of triterpenoids (**5** vs. **6**; **9** vs. **10**). Further, the different configuration of C-24 displayed a strong effect on their activities (24*S* > 24*R*, **6** vs. **7**).

Moreover, a dose-dependent experiment was conducted for compounds **6**, **7**, **10** and **16** at the concentration of 10, 20, and 40 μM, respectively. Consequently, **6**, **7**, as well as **16** were found to inhibit NO release from RAW264.7 cells in a dose-dependent manner (Figure 5).

As we referred to in the Introduction, IL-6, NF-κB, and iNOS are the major inflammatory cytokines, the Western blot method was used to study the anti-inflamatory machanism of compounds **6**, **7**, **10,** and **16** by determining their expressions in LPS-induced RAW264.7 cells. Comparing with the normal group, LPS led an obvious upregulation in the protein expressions of IL-6, NF-κB, and iNOS. Compounds **6**, **7**, **10**, and **16** could inhibit the protein expressions of IL-6, NF-κB, and iNOS in the cells. And the activities of **6**, **7**, and **16** were found to exhibit in a dose-dependent manner (Figure 6, Figure 7, Figure 8 and Figure 9).

## 3. Experimental

### 3.1. Experimental Procedures for Phytochmistry Study

#### 3.1.1. General Experimental Procedures

NMR spectra were determined on a Bruker ascend 600 MHz and/or Bruker ascend 500 MHz NMR spectrometer (Bruker BioSpin AG Industriestrasse 26 CH-8117, Fällanden, Switzerland) (internal standard: TMS). Positive- and Negative-ion mode HRESI-TOF-MS were measured on an Agilent Technologies 6520 Accurate-Mass Q-Tof LC/MS spectrometer (Agilent Corp., Santa Clara, CA, USA). Optical rotations, UV and IR spectra were run on a Rudolph Autopol^®^ IV automatic polarimeter (l = 50 mm) (Rudolph Research Analytical, Hackettstown NJ, USA), Varian Cary 50 UV-Vis (Varian, Inc., Hubbardsdon, MA, USA) and Varian 640-IR FT-IR spectrophotometer (Varian Australia Pty Ltd., Mulgrave, Australia), respectively.

CC were performed over macroporous resin D101 (Haiguang Chemical Co., Ltd., Tianjin, China), silica gel (48–75 μm, Qingdao Haiyang Chemical Co., Ltd., Qingdao, China), ODS (50 μm, YMC Co., Ltd., Tokyo, Japan), and Sephadex LH-20 (Ge Healthcare Bio-Sciences, Uppsala, Sweden). High performance liquid chromatography (HPLC) column: Cosmosil 5C_18_-MS-II (4.6 mm i.d. × 250 mm, 5 µm) and Cosmosil 5C_18_-MS-II (20 mm i.d. × 250 mm, 5 µm, Nakalai Tesque, Inc., Tokyo, Japan), Cosmosil PBr (4.6 mm i.d. × 250 mm, 5 µm) and Cosmosil PBr (20 mm i.d. × 250 mm, Nacalai Tesque, Inc., Kyoto, Japan), Cosmosil 5SL-II (4.6 mm i.d. × 250 mm, 5 µm) and 5SL-II (20 mm i.d. × 250 mm, 5 µm, Nakalai Tesque, Inc., Tokyo, Japan), and Venusil PrepG C18 (50 mm i.d. × 250 mm, 10 µm, Agela technologies, Tianjin, China) were used to analysis and separate the constituents.

#### 3.1.2. Plant Material

The roots of *Eurycoma longifolia* Jack were collected from the Nuang Mountain Recreation Forest in Selangor city, Malaysia, and identified by Dr. Wang Tao (Institute of Traditional Chinese Medicine, Tianjin University of Traditional Chinese Medicine). The voucher specimen was deposited at the Academy of Traditional Chinese Medicine of Tianjin University of TCM.

#### 3.1.3. Extraction and Isolation

The dried roots of *E. longifolia* (3.4 kg) were cut to pieces and refluxed with 70% EtOH to gain 70% EtOH extract (160.0 g, EL). Then, EL (125.0 g) was partitioned in an EtOAc-H_2_O mixture (1:1, *v/v*) to obtain EtOAc layer (ELE, 35.5 g) and H_2_O layer (85.0 g), respectively. The H_2_O layer was subjected to D101 macroporous resin CC (H_2_O → 95% EtOH). As a result, H_2_O (50.8 g) and 95% EtOH (31.3 g) eluates were obtained.

The 95% EtOH eluates (25.0 g) was subjected to ODS CC [MeOH-H_2_O (10:90 → 20:80 → 30:70 → 40:60 → 50:50 → 60:40 → 100:0, *v/v*)], and 19 fractions (ELG1–ELG19) were yielded. ELG2 (779.5 mg) was separated by pHPLC [CH_3_CN-1% HAc (10:90, *v/v*), Cosmosil PBr column], and ten fractions (ELG2-1–ELG2-10) were given. ELG2-8 (42.3 mg) was isolated by pHPLC [CH_3_CN-1% HAc (10:90, *v/v*), Cosmosil PBr column] to yield 3-chloro-4-hydroxyl benzoic acid-4-*O*-β-d-glucopyranoside (**14**, 15.7 mg). ELG3 (410.1 mg) was purified by pHPLC [CH_3_CN-1% HAc (8:92, *v/v*), Cosmosil PBr column] to gain isotachioside (**15**, 13.9 mg). ELG9 (3.2 g) was subjected to pHPLC [CH_3_CN-1% HAc (12:88, *v/v*), Venusil PrepG C18 column] to produce eight fractions (ELG9-1–ELG9-8). ELG9-4 (923.6 mg) was purified by pHPLC [CH_3_CN-1% HAc (9:91, *v/v*), Cosmosil PBr column] to yield eurylophenoloside A (**1**, 7.9 mg). ELG15 (1490.0 mg) was subjected to Sephadex LH-20 CC [MeOH-H_2_O (1:1, *v/v*)], and five fractions (ELG15-1–ELG15-5). ELG15-4 (283.3 mg) was isolated by pHPLC [CH_3_CN-1% HAc (17:83, *v/v*), Cosmosil PBr column] to yield eurylolignanoside A (**3**, 10.2 mg). ELG15-5 was purified by pHPLC [CH_3_CN-1% HAc (19:81, *v/v*), Cosmosil PBr column] to produce eurylolignanoside B (**4**, 16.3 mg). ELG16 (841.3 mg) was separated by Sephadex LH-20 CC [MeOH-H_2_O (1:1, *v/v*)] to give two fractions (ELG16-1–ELG16-2). ELG16-2 (633.4 mg) was isolated by pHPLC [CH_3_CN-1% HAc (22:78, *v/v*), Cosmosil PBr column] to yield six fractions (ELG16-2-1–ELG16-2-6). ELG16-2-3 (33.7 mg) was further purified by pHPLC [CH_3_CN-1% Hac (22:78, *v/v*), Cosmosil 5C_18_-MS-II column] to gain eurylophenoloside B (**2**, 5.0 mg).

ELE (25.0 g) was subjected to silica gel CC [CH_2_Cl_2_-EtOAc (50:1 → 20:1 → 5:1 → 2:1 → 1:1 → 1:20 → 1:50 → 0:100, *v/v*) → MeOH → MeOH + NH_3_·H_2_O] to produce thirteen fractions (ELE-1–ELE-13). ELE-6 (2.0 g) was separated by silica gel CC [PE-EtOAc (100:0 → 98:2 → 94:6 → 90:10 → 88:12 → 84:16 → 76:24 → 72:28 → 0:100, *v/v*)], and twelve fractions (ELE6-1–ELE6-12). ELE6-8 (102.6 mg) was purified by pHPLC [*n*-hexane-EtOAc (3:1, *v/v*), Cosmosil 5SL-II column] to give bourjotinolone B (**8**, 11.3 mg). Using the same separation condition, bourjotinolone A (**10**, 60.3 mg) was obtained from ELE6-10 (179.6 mg). ELE-7 (2.2 g) was subjected to silica gel CC [CH_2_Cl_2_-EtOAc (100:1 → 100:3 → 100:5 → 100:7 → 10:1 → 5:1 → 10:3 → 2:1 → 1:1 → 0:1, *v/v*) → MeOH] to yield twenty-one fractions (ELE7-1–ELE7-21). ELE7-4 (62.1 mg) was isolated by pHPLC [CH_2_Cl_2_-EtOAc (15:1, *v/v*), Cosmosil 5SL-II column] to give syringaldehyde (**12**, 3.9 mg) and scopoletin (**16**, 16.0 mg). ELE7-7 (81.9 mg) was separated by pHPLC [CH_3_CN-1% HAc (23:77, *v/v*), Cosmosil 5C_18_-MS-II column] to yield 3-methoxy-4-hydroxybenzoic acid (**11**, 10.1 mg) and 3-chloro-4-hydroxybenzoic acid (**13**, 5.3 mg). ELE7-13 (131.8 mg) was purified by pHPLC [CH_2_Cl_2_-MeOH (100:2, *v/v*), Cosmosil 5SL-II column] to give piscidinol A (**6**, 15.3 mg) and 3-episapeline A (**9**, 6.0 mg). ELE7-14 (76.2 mg) was isolated by pHPLC [CH_2_Cl_2_-MeOH (100:2, *v/v*), Cosmosil 5SL-II column] to gain hispidol B (**5**, 2.6 mg) and 24-*epi*-piscidinol A (**7**, 2.7 mg).

*Eurylophenoloside A (**1**):* White powder; [α]_D_^25^ −79.0 (*c* 0.20, MeOH); UV λ_max_ (MeOH) nm (log *ε*) 250 (3.91), 256 (3.94), 263 (3.81); IR (KBr) *υ*_max_ 3374, 2935, 2884, 1601, 1505, 1462, 1421, 1228, 1197, 1126, 1067 cm^−1^; ^1^H NMR (C_5_D_5_N, 600 MHz) and ^13^C NMR (C_5_D_5_N, 150 MHz) see Table 1. HRESI-TOF-MS *m/z* 633.2005 [M + Na]^+^ (calcd for C_25_H_38_O_17_Na, 633.2001).

*Eurylophenoloside B (**2**):* White powder; [α]_D_^25^ −34.7 (*c* 0.15, MeOH); UV λ_max_ (MeOH) nm (log *ε*): 222 (4.29, sh), 278 (3.85); IR (KBr) υ_m__ax_ 3357, 2932, 2886, 2837, 1704, 1603, 1506, 1462, 1423, 1379, 1335, 1275, 1225, 1129, 1076, 1041 cm^−1^; ^1^H NMR (C_5_D_5_N, 500 MHz) and ^13^C NMR (C_5_D_5_N, 125 MHz) see Table 2. HRESI-TOF-MS *m/z* 813.2438 [M + H]^+^ (calcd for C_34_H_47_O_21_, 813.2424).

*Eurylolignanoside A (**3**):* White powder; [α]_D_^25^ −47.2 (*c* 0.25, MeOH); UV λ_max_ (MeOH) nm (log *ε*): 217 (4.26, sh), 267 (3.86); CD (conc 0.007 M, MeOH) mdeg (*λ*nm): −62.6 (225 nm), −23.1 (259 nm); IR υ_max_ (KBr): 3382, 2923, 2881, 1712, 1586, 1515, 1461, 1423, 1376, 1314, 1273, 1228, 1154, 1073, 1040 cm^−1^; ^1^H NMR (CD_3_OD, 500 MHz) and ^13^C NMR (CD_3_OD, 125 MHz) see Table 3. HRESI-TOF-MS *m/z* 707.2521 [M + Na]^+^ (calcd for C_32_H_44_O_16_Na, 707.2522).

*Eurylolignanoside B (**4**):* White powder; [α]_D_^25^ −42.5 (*c* 0.40, MeOH); UV λ_max_ (MeOH) nm (log *ε*) 232 (3.71, sh), 284 (3.41); CD (conc 0.008 M, MeOH) mdeg (λnm): −15.1 (231 nm), −4.2 (291 nm); IR (KBr) *υ*_max_ 3408, 2935, 2851, 1702, 1600, 1557, 1516, 1460, 1423, 1390, 1273, 1211, 1152, 1070, 1031 cm^−1^; ^1^H NMR (CD_3_OD, 500 MHz) and ^13^C NMR (CD_3_OD, 125 MHz) see Table 4. HRESI-TOF-MS *m/z* 509.1664 [M − H]^−^ (calcd for C_24_H_29_O_12_, 509.1666).

*3-Chloro-4-hydroxyl benzoic acid-4-O-β**-d-glucopyranoside* (**14**)*:* White powder; ^1^H NMR (C_5_D_5_N, 500 MHz): δ 8.47 (1H, d, *J* = 1.5 Hz, H-2), 7.67 (1H, d, *J* = 8.5 Hz, H-5), 8.21 (1H, dd, *J* = 1.5, 8.5 Hz, H-6), 5.80 (1H, d, *J* = 7.5 Hz, H-1’), 4.38 (1H, m, overlapped, H-2’), 4.37 (1H, m, overlapped, H-3’), 4.32 (1H, dd, *J* = 8.5, 8.5 Hz, H-4’), 4.15 (1H, m, H-5’), [4.38 (1H, m, overlapped), 4.53 (1H, dd, *J* = 2.5, 12.0 Hz), H_2_-6’]; ^13^C NMR (C_5_D_5_N, 125 MHz): δ 127.1 (C-1), 132.2 (C-2), 123.0 (C-3), 157.1 (C-4), 115.9 (C-5), 130.4 (C-6), 167.8 (C-7), 101.8 (C-1’), 74.7 (C-2’), 78.6 (C-3’), 71.0 (C-4’), 79.2 (C-5’), 62.3 (C-6’) data were firstly reported here; HRESI-TOF-MS *m/z* 333.0383 [M − H]^−^ (calcd for C_13_H_14_ClO_8_, 333.0389).

Acid hydrolysis of **1**–**4***:* The acid hydrolysis reactions of **1**–**4** were conducted by using the method reported previously [24]. As a result, d-glucose (12.4 min, positive optical rotation) for **1**–**4** was identified by a comparison of their retention time and optical rotation with that of an authentic sample.

### 3.2. Experimental Procedures for Bioassay

#### 3.2.1. General Experimental Procedures

MTT and nitrite levels were measured on a BioTek Cytation five-cell imaging multi-mode reader (Winooski, VT, USA); protein bands were mixed with Enhanced Chemiluminescence (Millipore, Billerica, MA, USA); protein bands were visualized with the Amersham imager 600 luminescent image analyzer (GE healthcare Japan Co., Tokyo, Japan).

RAW264.7 cells were obtained from the cell center at the Chinese Academy of Medical Science; LPS and Dex were purchased from Sigma Chemical (St. Louise, MO, USA); penicillin and streptomycin were purchased from Thermo Fisher Scientific (Waltham, MA); dulbecco’s modified eagle medium (DMEM) and fetal bovine serum (FBS) were purchased from Biological Industries (Beit Haemek, IN); nitric oxide fluorometric assay kit was purchased from Beyotime Biotechnology (Shanghai, China); bicinchoninic acid protein assay kit was purchased from Thermo Fisher Scientific (Waltham, MA, USA). Rabbit anti-IL-6 was purchased from Proteintech Group, Inc (Chicago, IL, USA). Rabbit anti-NF-κB, iNOS, and β-actin were purchased from Abcam plc. (Cambridge, MA, USA). Horseradish peroxidase-conjugated anti-rabbit immunoglobulin G (IgG) was purchased from Zhongshan Goldbridge Biotechnology (Beijing, China).

#### 3.2.2. Cell Culture

RAW 264.7 cells were maintained in high glucose DMEM supplemented with 10% heat-inactivated FBS, 100 U/mL penicillin, and 100 µg/mL streptomycin in a humidified atmosphere containing 5% CO_2_ at 37 °C. Cells were grown to 80% confluence and then seeded at 2 × 10^6^ cells/mL density in 24-well plates incubated before treatment.

#### 3.2.3. Cell Viability Assay

MTT colorimetric assay was used to determine cell viability. In brief, RAW 264.7 cells were seeded in 24-well plastic plates and treated without or with test samples (40 μM) for 24 h, respectively. The culture condition was the same as 3.2.2. The medium was removed, and the cells were incubated with 0.5 mg/mL of MTT solution. After 4 h incubation, the supernatant was removed and formation of formazan. The absorbance at 490 nm was measured with a microplate reader.

#### 3.2.4. Measurement of NO levels

Initially, 2 × 10^6^ cells/mL RAW264.7 cells were seeded on a 24-well plate and incubated overnight. After 24h, the media were changed, which contained LPS (0.5 μg/mL) with or without tested compounds **1**–**16** (40 μM), as well as the positive drug DEX (1 μg/mL), and then, the cells were incubated for 24 h. The cell supernatant was collected to detect NO levels, and cells were harvested for protein analysis.

#### 3.2.5. Western Blot Analysis

As described previously [28], the RAW264.7 cells treated with LPS were analyzed by Western blot. The protein concentrations in the supernatants and tissues were quantified using a bicinchoninic acid protein assay kit. Firstly, 60 μg of protein was mixed with 4 × loading dye (Laemmli Buffer) and 2-mercapto ethanol, before being heated at 100 °C for 5 min. The protein was resolved by 10% sodium dodecyl sulfate polyacrylamide gel electrophoresis and transferred to immunoblot polyvinylidene difluoride (PVDF) membranes (Merck Millipore Ltd., Darmstadt, Germany). The membranes were incubated at 4 °C overnight with primary antibodies against IL-6 (1:1000) (21865-1-AP, Proteintech), NF-κB (1:500) (ab16502, Abcam), iNOS (1:1000) (ab3523, Abcam), and β-actin (1:1000) (ab8227, Abcam). Then, the membranes were washed three times with Tris-buffered saline/Tween 20 (TBS-T; 10 min each time) and incubated with a horseradish peroxidase-labeled secondary goat anti-rabbit (1:10,000) antibody for 1 h at room temperature. Next, the blots were again washed three times with TBS-T (10 min each time).

Finally, protein bands were mixed with enhanced chemiluminescence (Millipore Co., Ltd., MA, USA). Subsequently, the protein bands were visualized with the Amersham imager 600 luminescent image analyzer. The intensities of protein bands were quantified by Image J analysis software.

#### 3.2.6. Statistical Analysis

Values are expressed as mean ± S.D. SPSS 17.0 was used to conduct the statistics of all the grouped data. *p* < 0.05 was considered to indicate statistical significance. One-way analysis of variance (ANOVA) and Tukey’s studentized range test were used for the evaluation of the significant differences between means and post hoc, respectively.

## 4. Conclusions

In summary, four new phenolic acids, eurylophenolosides A (**1**) and B (**2**), eurylolignanosides A (**3**) and B (**4**), along with twelve known isolates were obtained from the 70% ethanol extract of *E. longifolia* roots by a combination of various chromatographic methods and spectral techniques. Among the known compounds, **12**–**15** were isolated from the *Eurycoma* genus for the first time. Furthermore, the NMR data of **14** was firstly reported here.

Compounds **6**, **7**, **10**, and **16** showed significant inhibitory activity on NO release from RAW264.7 cells. In order to make the mechanism of their anti-inflammatory activities clear, Western blot assays were conducted. As a result, all of them were found to inhibit the LPS-induced protein expression of IL-6, NF-κB, and iNOS in the NF-κB signaling pathway. Moreover, the protein expression inhibitory effects of **6**, **7**, and **16** were found to be in a dose-dependent manner. The mechanism may be related to inhibiting the IL-6-induced NF-κB pathway to suppress iNOS expressions. The experiment suggested terpenes and coumarins might also be the anti-inflammatory substances in *E. longifolia* roots, except for alkaloids [13].

On the other hand, acording to references, compound **16** possessed the ability to down-regulate the protein expression level of IL-6 and NF-κB [29], as well as the gene expression level of iNOS [30], which was identical to our experimental results. Meanwhile, although compounds **6**, **7,** and **10** have been reported to possess in vitro anti-inflammatory activities [31,32], their anti-inflammatory mechanisms have not yet been reported. Thus, our investigation not only verified the research on the anti-inflammatory mechanism of compound **16**, but also complemented the blank of exisiting literature on the anti-inflammation mechanisms of compounds **6**, **7**, and **10**. Furtherly, the strong inhibitory activities of compound **6** indicated that it could be the drug candidates for inflammation-related diseases.

## Figures and Tables

**Figure 1 molecules-24-03157-f001:**
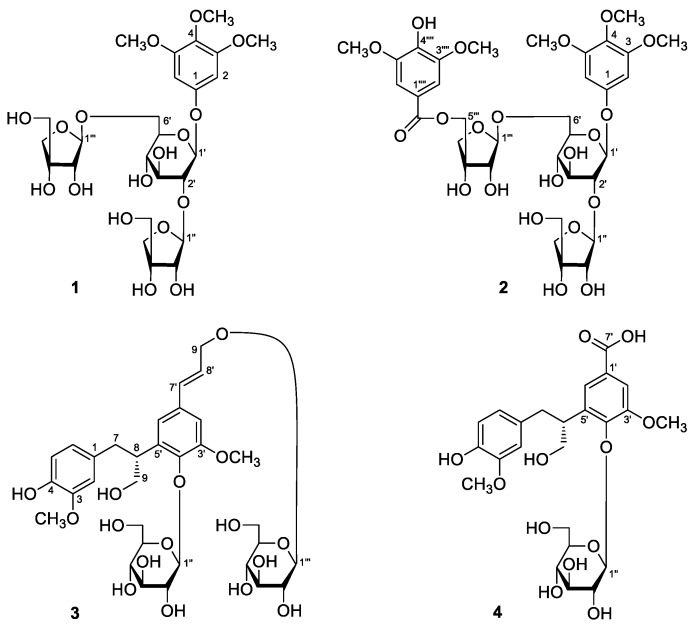
The new compounds **1**–**4** obtained from *E. longifolia* roots.

**Figure 2 molecules-24-03157-f002:**
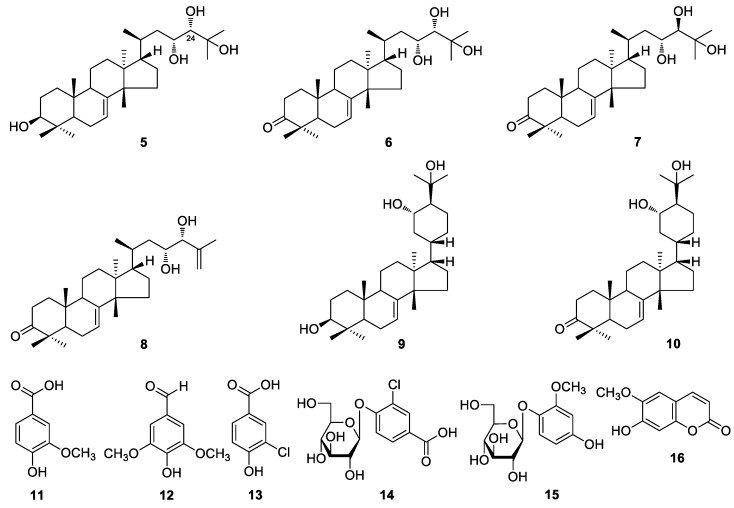
The known compounds **5**–**16** obtained from *E. longifolia* roots.

**Figure 3 molecules-24-03157-f003:**
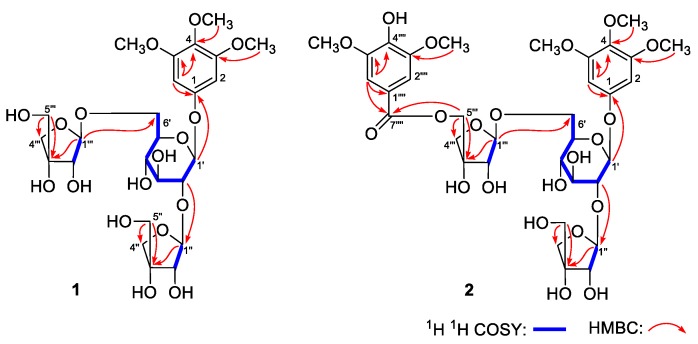
The main ^1^H ^1^H COSY and HMBC correlations of **1** and **2.**

**Figure 4 molecules-24-03157-f004:**
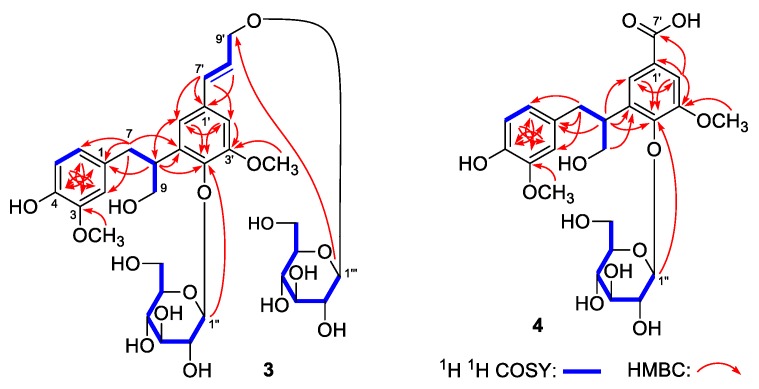
The main ^1^H ^1^H COSY and HMBC correlations of **3** and **4.**

**Figure 5 molecules-24-03157-f005:**
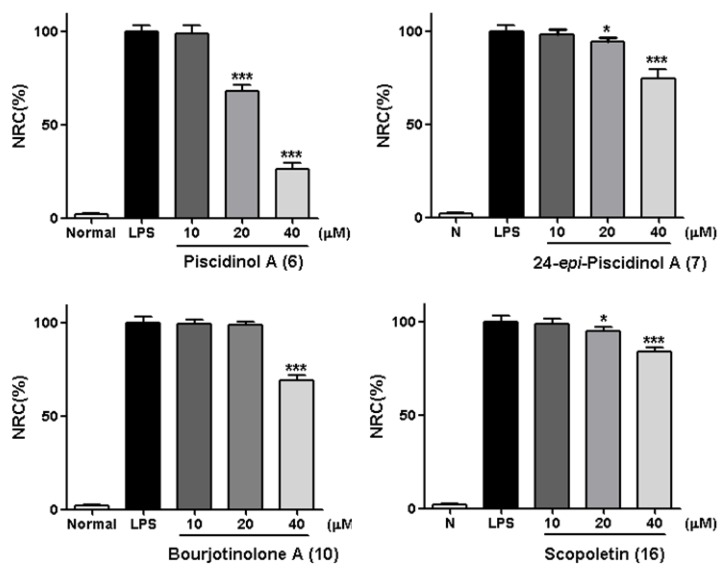
Inhibitory effects of compounds **6**, **7**, **10** and **16** at concentration of 10, 20, and 40 μM on NO production in RAW 264.7 cells, respectively. Normal: the normal group without LPS and other tested samples. Nitrite relative concentration (NRC): percentage of control groupn (set as 100%). Values represent the mean ± SD of four determinations. * *p* < 0.05; *** *p* < 0.001 (Differences between compound-treated group and control group). *n* = 4.

**Figure 6 molecules-24-03157-f006:**
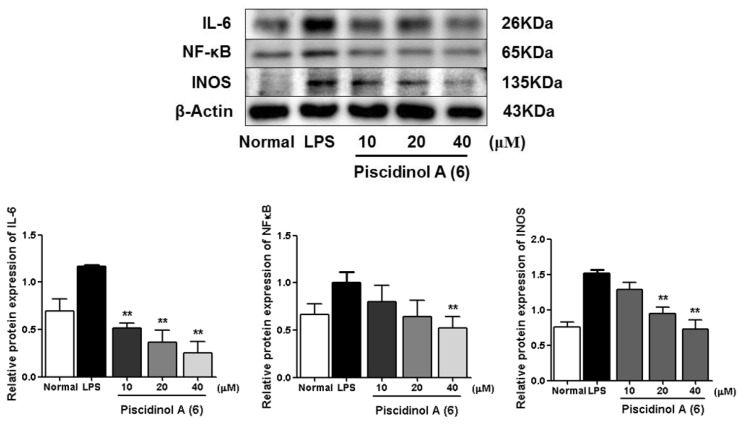
Inhibitory effects of compound **6** on the protein expression of IL-6, NF-κB and iNOS in RAW 264.7 cells. Normal: normal group without LPS, DEX and other tested samples. Values represent the mean ± SEM of three determinations. ** *p* < 0.01; (Differences between compound-treated group and control group). *n* = 3.

**Figure 7 molecules-24-03157-f007:**
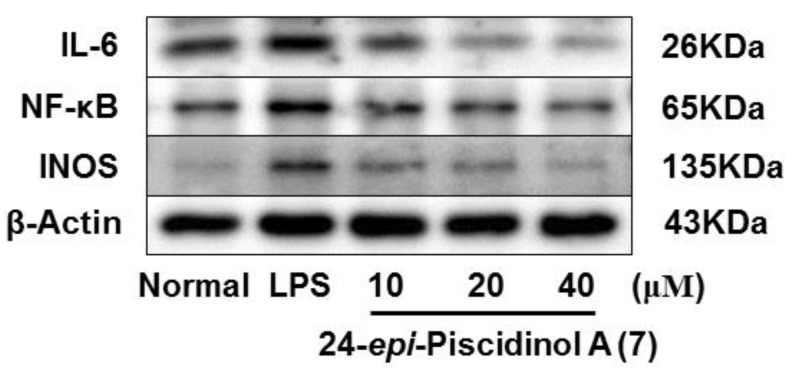
Inhibitory effects of compound **7** on the protein expression of IL-6, NF-κB and iNOS in RAW 264.7 cells. Normal: normal group without LPS, DEX and other tested samples. Values represent the mean ± SEM of three determinations. * *p* < 0.05 (Differences between compound-treated group and control group). *n* = 3.

**Figure 8 molecules-24-03157-f008:**
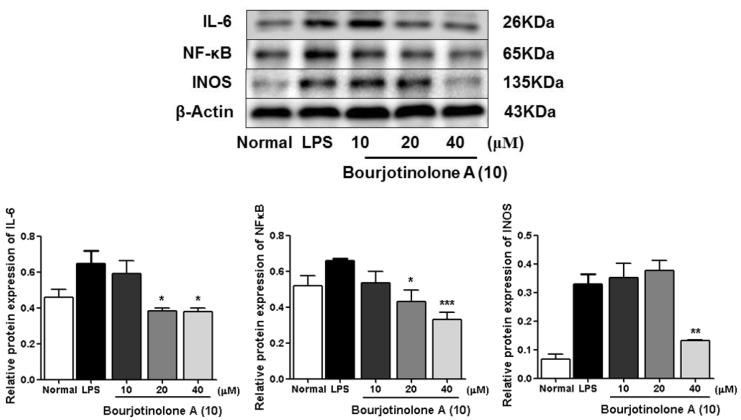
Inhibitory effects of compound **10** on the protein expression of IL-6, NF-κB and iNOS in RAW 264.7 cells. Normal: normal group without LPS, DEX and other tested samples. Values represent the mean ± SEM of three determinations. * *p* < 0.05; ** *p* < 0.01; *** *p* < 0.001 (Differences between compound-treated group and control group). *n* = 3.

**Figure 9 molecules-24-03157-f009:**
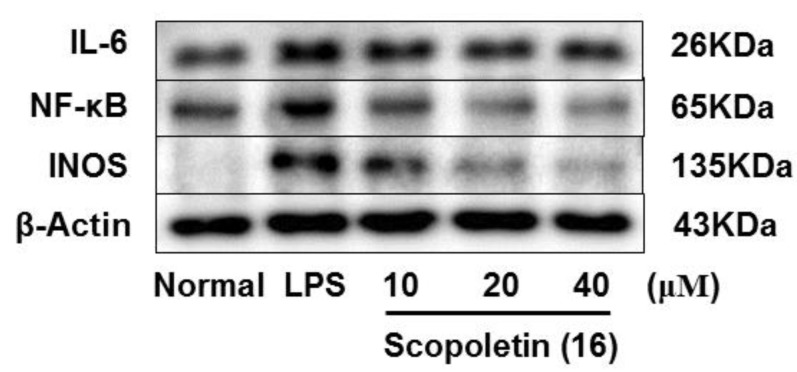
Inhibitory effects of compound **16** on the protein expression of IL-6, NF-κB and iNOS in RAW 264.7 cells. Normal: normal group without LPS, DEX and other tested samples. Values represent the mean ± SEM of three determinations. * *p* < 0.05; ** *p* < 0.01; *** *p* < 0.001 (Differences between compound-treated group and control group). *n* = 3.

**Table 1 molecules-24-03157-t001:** ^1^H and ^13^C NMR data of **1** in C_5_D_5_N.

No.	δ_C_	δ_H_ (*J* in Hz)	No.	δ_C_	δ_H_ (*J* in Hz)
1	155.5	-	4’’	75.9	4.47 (d, 9.6)
2,6	95.9	6.94 (s)			4.93 (d, 9.6)
3,5	154.4	-	5’’	66.6	4.25 (d, 11.4)
4	134.3	-			4.28 (d, 11.4)
1’	101.9	5.38 (d, 7.8)	1’’’	111.0	5.66 (d, 2.4)
2’	77.1	4.50 (dd, 8.4, 7.8)	2’’’	77.6	4.69 (d, 2.4)
3’	79.0	4.31 (dd, 9.0, 8.4)	3’’’	80.3	-
4’	71.8	3.95 (dd, 9.0, 9.0)	4’’’	74.9	4.31 (d, 9.6)
5’	77.2	4.18 (m)			4.55 (d, 9.6)
6’	69.0	4.03 (dd, 10.8, 7.2)	5’’’	65.2	4.09 (d, 11.4)
		4.78 (br. d, ca. 11)			4.14 (d, 11.4)
1’’	110.5	6.60 (br. s)	3,5-OCH_3_	56.2	3.88 (s)
2’’	78.0	4.77 (br. s)	4-OCH_3_	60.6	3.79 (s)
3’’	81.0	-			

**Table 2 molecules-24-03157-t002:** ^1^H and ^13^C NMR data of **2** in C_5_D_5_N.

No.	δ_C_	δ_H_ (*J* in Hz)	No.	δ_C_	δ_H_ (*J* in Hz)
1	155.4	-	1’’’	110.6	5.69 (d, 3.0)
2,6	96.0	6.92 (s)	2’’’	78.6	4.62 (d, 3.0)
3,5	154.3	-	3’’’	78.7	-
4	134.1	-	4’’’	74.8	4.39 (d, 9.5)
1’	101.9	5.40 (d, 8.0)			4.49 (d, 9.5)
2’	77.2	4.49 (dd, 9.0, 8.0)	5’’’	67.8	4.86 (d, 11.5)
3’	78.9	4.32 (dd, 9.0, 9.0)			4.90 (d, 11.5)
4’	71.7	3.98 (dd, 9.5, 9.0)	1’’’’	120.1	-
5’	77.1	4.17 (m)	2’’’’	108.3	7.68 (s)
6’	68.9	4.06 (dd, 11.5, 7.5)	3’’’’	148.7	-
		4.76 (br. d, ca. 12)	4’’’’	143.0	-
1’’	110.5	6.57 (br. s)	5’’’’	148.7	-
2’’	78.1	4.78 (br. s)	6’’’’	108.3	7.68 (s)
3’’	81.0	-	7’’’’	166.7	-
4’’	75.8	4.46 (d, 9.5)	3,5-OCH_3_	56.2	3.87 (s)
		4.91 (d, 9.5)	4-OCH_3_	60.6	3.79 (s)
5’’	66.5	4.24 (d, 11.0)	3’’’’,5’’’’-OCH_3_	56.3	3.77 (s)
		4.27 (d, 11.0)			

**Table 3 molecules-24-03157-t003:** ^1^H and ^13^C NMR data of **3** in CD_3_OD.

No.	δ_C_	δ_H_ (*J* in Hz)	No.	δ_C_	δ_H_ (*J* in Hz)
1	133.2	-	9’	70.8	4.33 (ddd, 13.0, 6.0, 1.0)
2	113.8	6.57 (d, 1.5)			4.52 (ddd, 13.0, 6.0, 1.0)
3	148.4	-	1’’	105.4	4.68 (d, 7.5)
4	145.4	-	2’’	76.0	3.46 (dd, 8.0, 7.5)
5	115.7	6.56 (d, 8.0)	3’’	78.0	3.41 (dd,8.0, 8.0)
6	122.6	6.48 (dd, 8.0, 1.5)	4’’	71.3	3.37 (dd, 8.5, 8.5)
7	39.2	2.73 (dd, 14.0, 9.5)	5’’	77.9	3.12 (m)
		2.96 (dd, 14.0, 5.5)	6’’	62.5	3.66 (m, overlapped)
8	42.8	3.97 (m)			3.77 (m, overlapped)
9	66.8	3.69 (m, overlapped)	1’’’	103.3	4.37 (d, 8.0)
		3.76 (m, overlapped)	2’’’	75.2	3.24 (dd, 8.5, 8.0)
1’	135.2	-	3’’’	78.2	3.37 (dd, 8.5, 8.5)
2’	109.2	6.95 (d, 2.0)	4’’’	71.7	3.29 (m, overlapped)
3’	153.5	-	5’’’	78.1	3.29 (m, overlapped)
4’	145.2	-	6’’’	62.9	3.68 (m, overlapped)
5’	139.0	-			3.88 (dd, 12.0, 2.0)
6’	119.4	6.92 (d, 2.0)	3-OCH_3_	56.3	3.69 (s)
7’	133.7	6.65 (br. d, ca. 16.0)	3’-OCH_3_	56.4	3.83 (s)
8’	126.4	6.30 (dt, 16.0, 6.0)			

**Table 4 molecules-24-03157-t004:** ^1^H and ^13^C NMR data of **4** in CD_3_OD.

No.	δ_C_	δ_H_ (*J* in Hz)	No.	δ_C_	δ_H_ (*J* in Hz)
1	133.1	-	4’	148.8	-
2	113.8	6.60 (d, 1.5)	5’	139.0	-
3	148.5	-	6’	122.6	7.63 (br. s)
4	145.4	-	7’	170.4	-
5	115.8	6.57 (d, 8.0)	1’’	105.0	4.80 (d, 8.0)
6	122.6	6.49 (dd, 8.0, 1.5)	2’’	76.0	3.48 (dd, 8.0, 8.0)
7	39.3	2.73 (dd, 14.0, 9.5)	3’’	77.9	3.43 (dd, 9.0, 8.0)
		2.99 (dd, 14.0, 6.0)	4’’	71.2	3.39 (dd, 8.5, 8.5)
8	43.0	3.99 (m)	5’’	78.1	3.13 (m)
9	66.7	3.71 (m)	6’’	62.4	3.66 (dd, 12.0, 5.0)
		3.79 (m)			3.75 (dd, 12.0, 2.0)
1’	148.5	-	3-OCH_3_	56.3	3.71 (s)
2’	112.5	7.49 (br. s)	3’-OCH_3_	56.4	3.85 (s)
3’	153.2	-			

**Table 5 molecules-24-03157-t005:** Inhibitory effects of compounds **1**–**16** on NO production in RAW 264.7 cells.

NO.	NRC (%)	NO.	NRC (%)
Normal	2.2 ± 0.4	8	101.3 ± 6.1
Control	100 ± 3.5	9	90.3 ± 5.3 *
DEX	82.7 ± 3.1 ***	10	69.0 ± 2.7 ***
1	93.6 ± 3.7 *	11	98.5 ± 2.8
2	101.1 ± 3.6	12	96.3 ± 2.4
3	102.2 ± 4.0	13	95.1 ± 0.9
4	96.0 ± 1.1	14	98.2 ± 5.2
5	87.0 ± 3.3 *	15	97.1 ± 1.0
6	26.5 ± 3.0 ***	16	83.9 ± 2.2 ***
7	74.9 ± 4.4 ***		

Normal: normal group without LPS, DEX and other tested samples. Control: lipopolysaccharide (LPS). Positive control: Dexamethasone (Dex). Nitrite relative concentration (NRC): percentage of control groupn (set as 100%). Values represent the mean ± SD of three determinations. * *p* < 0.05; *** *p* < 0.001 (Differences between compound-treated group and control group). *n* = 4. Final concentration was 40 μM for **1–16**, was 1.0 μg/mL for positive control (Dex), respectively.

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
