# Peer review of "Bioactive Constituents from the Roots of Eurycoma longifolia"

_molecules, 2019, doi:10.3390/molecules24173157_

Round 1

Reviewer 1 Report

In the present study, the chemical composition of the roots of E. longifolia was studied. Totally sixteen compounds were identified, including four new phenolic components were characterized by using various spectral techniques and chemical reactions. In addition, the anti-inflammatory activity and mechanism of some isolated compounds were discussed. Although I appreciated authors′ efforts on purifying these new compounds and studying the bioactivity, the present research work exhibited some technical deficiencies and it should be reconsidered the acceptance after authors′ major revisions. Therefore, this manuscript is not recommended to accept for publication in Molecules in the present form. In addition, there were some major comments addressed as following.

The English language and style of this manuscript were well edited; however, there were still some minor typographic, grammar, and format errors to be observed. Authors have to check and revise these errors. For examples, line 121, the trivial name of compound 1; the order of coupling constants in the text and Tables; the significant figures of spectral data in the Experimental section should be uniform, etc. Syringaldehyde was already reported in the following paper, and authors have to include this article in the reference. Kuo et al., Med. Chem. 2004, 12, 537-544. The configuration and conformation of apiofuranose moieties in compounds 1 and 2 should be further verified. According to the coupling constants of anomeric protons, the β conformation was questionable. The HRMS of compound 2 was deviated from the theoretical values a little more. The configurations of C-8 in compounds 3 and 4 should be supported by more spectral evidences. In addition, the present drawing was not R There were not any canthin-6-one and quassinoid derivatives reported in the present study. Authors have to rationalize this point. According to the experimental data in Table 5, the bioactivity of isolated compounds (40 μM) was not significant as compared with that of DEX (1 μg/mL). In my opinion, only compound 6 was a little interesting. In Figure 5, 10 was not dose-dependent, and 7 and 16 were not effectively inhibited NO production. In addition, more discussion of the mechanism study should be illustrated in the text. Some inhibitory results were not so strong as described in page 8 (lines 207-213). In the References section, the writing manner of some references did not follow the style of this journal. Authors have to check and revise these minor errors in this section.

Reviewer 2 Report

Concerns: 

1) This paper needs extensive grammatical editing by a native English speaker. 

2) In the introduction (p 1, line 45) you refer to RAW 264.7 cells as macrophages. This is incorrect. They are commonly used as a model for macrophages, but they are tumor cells. 

3) Triterpenoids and coumarins are very insoluble in aqueous solutions. You describe assaying them at 40 uM. Is this the limit of solubility? Otherwise, how do you know that this is the optimal concentration for assay? 

4) Further description of the MTT assay is required. 

5) Further description of the NO assay is required. 

6) In description of your bioassay results, the word 'normal' is ambiguous and should be avoided. 

7) The bioassay results seem to lack proper controls for cross reaction of the natural products with the assay ingredients (e.g. compound specific controls). 

8) When you present controls, you must clarify what they are (e.g. MTT and NO assay). 

9) Triterpenoids and coumarins are both poorly water soluble. What means are you using to dissolve them for your bioassays? Are you also checking for residue and precipitate on the plates after the assay? 

10) Please consider providing raw quantification data for the western blot analysis in the supplemental. The graphs do not appear to match the blots for many of the assays. 

Round 2

Reviewer 1 Report

The revised manuscript is acceptable. Thus, the reviewer recommend it for publication. 

Reviewer 2 Report

Thank you for your detailed responses to my comments.